# Long-Term Antifogging Coating Based on Black Phosphorus Hybrid Super-Hydrophilic Polymer Hetero-Network

**DOI:** 10.3390/nano13010086

**Published:** 2022-12-24

**Authors:** Lie Wu, Yihong Kang, Yuhao Deng, Fan Yang, Rui He, Xue-Feng Yu

**Affiliations:** 1Materials Interfaces Center, Shenzhen Institute of Advanced Technology, Chinese Academy of Sciences, Shenzhen 518055, China; 2University of Chinese Academy of Sciences, Beijing 100049, China; 3Hubei Three Gorges Laboratory, Yichang 443007, China

**Keywords:** black phosphorus, hybrid, polymer, hetero-network, coating, long-term antifogging

## Abstract

The antifogging coating based on super-hydrophilic polymer is regarded as the most promising strategy to avoid fogging but suffers from short-term effectiveness due to antifogging failure induced by water invasion. In this study, a black phosphorus nanosheets (BPs) hybrid polymer hetero-network coating (PUA/PAHS/BPs HN) was prepared by UV curing for the first time to achieve long-term antifogging performance. The polymer hetero-network (HN) structure was composed of two novel cross-linked acrylic resin and polyurethane acrylate. Different from physical blending, a covalent P-C bond between BPs and polymer is generated by UV initiated free radical reaction, resulting in BPs firmly embedded in the polymer HN structure. The BPs enriched on the coating surface by UV regulating migration prevent permeation of water towards the inside of the coating through its own good water-based lubricity and water absorption capacity. Compared with the nonhybrid polymer HN, PUA/PAHS/BPs HN not only has higher hardness and better friction resistance properties, but also exhibits superior water resistance and longer antifogging duration. Since water invasion was greatly reduced by BPs, the PUA/PAHS/BPs HN coating maintained antifogging duration for 60 min under a 60 °C water vapor test and still maintained long-term antifogging performance after being immersed in water for 5 days.

## 1. Introduction

When saturated water vapor condenses on a surface with a temperature lower than the dew point of the vapor, fog forms as tiny droplets that can scatter visible light [1,2,3]. For those transparent materials, surface fogging greatly reduces light transmittance, resulting in undesirable failure when applied in optical-related fields. For example, surface fog largely impacts the precision of optical and analytical instruments, such as infrared microscopes and clinical laparoscopy [4].

Many strategies, referring to super-hydrophobic and super-hydrophilic surfaces, have been developed for effective antifogging. Super-hydrophobic surfaces eliminate the effect of fog droplets by rolling down from the surface, but the light transmission tends to be compromised by the surface’s own micro/nano-structured roughness [4,5,6,7]. Different from super-hydrophobic surfaces, the antifogging coating based on super-hydrophilic polymer avoids fogging by quickly spreading fog droplets into a continuous water film, so as to effectively prevent the light scattering. Therefore, in the field of antifogging, super-hydrophilic surfaces are easier, more reliable and more promising than super-hydrophobic surfaces [8,9,10,11,12]. However, as the water film on a super-hydrophilic surface grows to a certain thickness, polymer-based coatings suffer from loss of water-soluble components and swelling-induced peeling and cracking, which results in the antifogging failure of the coatings [13]. Therefore, polymer-based coatings are usually limited to short-term antifogging effectiveness due to their insufficient water resistance capability. To improve the water resistance of polymer-based coatings, some water-resistant organics such as hydrophobic components are usually added into the coatings. Unfortunately, these organics usually weaken the hydrophilicity of the coating, resulting in a decline in its antifogging ability. To date, it is still a huge challenge to balance the hydrophilicity and water resistance of polymer-based coatings [14].

Generally, hydrophilic inorganic nanomaterials are able to absorb and store water without swelling, thus bringing a new strategy to regulate the balance between the hydrophilicity and water resistance of polymer-based coatings [15]. To improve the comprehensive properties of these organic–inorganic composites, more strategies for regulating compatibility, dispersion and stability of inorganic nanomaterials in the polymer-based coatings are required [16]. As a new hydrophilic two-dimensional inorganic nanomaterial [17,18], black phosphorus (BP) can improve the wear resistance and drainage capacity of the coating due to its excellent water-based lubrication performance [19,20,21]. On the other hand, BP possesses excellent compatibility with hydrophilic polymers, leading to universally good stability of the BP/polymer complex materials [22,23]. Unfortunately, the combination mode of BP with polymer reported at present is mostly weak physical blending rather than strong chemical combination [24,25,26,27], which greatly affects various performance attributes of BP-based polymer coatings. Therefore, although the BP-based polymer coating is a promising material for antifogging, some huge challenges, especially the chemical combination and microstructure regulation between BP and polymer, still need to be overcome.

In this study, a long-term antifogging coating based on BP nanosheets (BPs) hybrid super-hydrophilic polymer hetero-network (HN) was designed and prepared by construction of P-C bond under UV curing. The polymer hetero-network (HN) structure [28,29] was composed of two novel cross-linked compounds, acrylic resin and polyurethane acrylate. Different from physical blending, a covalent P-C bond between BPs and polymer is generated by UV-initiated free radical reaction, resulting in BPs firmly embedded in the polymer HN structure. The BPs enriched on the coating surface by UV regulating migration can effectively prevent the permeation of water towards the inside of the coating through its own good water-based lubricity and water absorption capacity. 

## 2. Materials and Methods

### 2.1. Materials

The BP crystals were obtained from Mophos (www.Mophos.cn, Yichang, China). Polyethylene glycol 2000 (PEG 2000), 4-methoxyphenol (MeHQ, AR, 99.0%), butylated hydroxytoluene (BHT), >99.0% (GC)), dibutyltin dilaurate (DBTDL, 95%), N-Methyl pyrrolidone (NMP), isophorone diisocyanate (IPDI, 99%), pentaerythritol triacrylate (PETA, 96%), ethyl acetate (99.0%, GC), acrylic acid (AA, >99.7%, GC), 2-hydroxyethyl methacrylate (HEMA, 99%), sulfobetaine methacrylate (SBMA), azodiisobutyronitrile (AIBN, 99%) and diphenyl (2, 4, 6-trimethylbenzoyl) phosphine oxide (TPO, 97%) were purchased from Aladdin Chemical Reagent Co., Ltd. (Shanghai China). All chemical regents were directly used without any further purification.

### 2.2. Instruments

The physicochemical properties of materials were characterized by scanning electron microscopy (SEM), atomic force microscopy (AFM), optical microscopy, Raman scattering microscopy, X-ray photoelectron spectroscopy (XPS), Fourier transform infrared (FTIR) spectroscopy and ^1^H nuclear magnetic resonance spectroscopy (^1^H NMR).

The mechanical properties of coatings were characterized by mechanical instruments. The adhesion strength, hardness and friction resistance of coatings were measured on a direct pull tensile force machine, pencil hardness tester and ball disc friction tester, respectively.

The hydrophilicity of coatings was measured by the water contact angle (WCA) tester, and the antifogging performance of the coatings was tested through a thermostatic water bath.

### 2.3. Methods

#### 2.3.1. Preparation of Materials

*Synthesis of polyurethane acrylate (PUA):* 2 g PEG 2000, 6 mg MeHQ, 3 mg BHT and 3 mg DBTDL were dissolved in 20 ml ethyl acetate to obtain the dropping solution. A 444 mg amount of IPDI was put into a three-port flask, and then the dropping solution was added to the three-port flask for 2 h at room temperature under the stirring speed of 200 rpm. The reaction was carried out at 40 °C for 6 h, and then 500 mg PETA was added to continue the reaction for 3 h at 80 °C. After filtration and solvent evaporation, the residue was dried in a vacuum to yield a colorless oil.

*Synthesis of P(AA-HEMA-SBMA) acrylic resin (PAHS):* In a three-port flask, 3.6 g AA, 6.5 g HEMA, 14 g SBMA and 24 mg AIBN were dissolved in 30 ml ethyl acetate, and the reaction was carried out at 80 °C for 6 h. After filtration and solvent evaporation, the residue was dried in a vacuum to yield a white powder.

*Synthesis of black phosphorus nanosheets (BPs):* The BPs were prepared by a liquid exfoliation method reported by our group. In brief, 10 mg of the bulk BP crystal was dispersed in 10 mL NMP and sonicated for 6 h with an ultrasonic frequency of 19–25 kHz (2 s ON and 4 s OFF; 1800 W; 6 °C). The dispersion was centrifuged for 15 min at 7000 rpm, and the collected supernatant was centrifuged for 15 min at 12000 rpm for further use.

*Synthesis of super-hydrophilic polymer hetero-network coating (PUA/PAHS HN):* The UV-curable solution was prepared by dissolving the photo initiator TPO and the obtained PUA, PAHS in 2-propanol with about 40% total solid content. The UV-curable solution was spin-coated on a plastic substrate, and then followed by UV irradiation (broadband, 400 mj/cm^2^) to obtain a cured PUA/PAHS/BPs HN coating.

*Synthesis of BPs hybrid super-hydrophilic polymer hetero-network coating (PUA/PAHS/BPs HN):* The UV-curable solution was prepared by dissolving the photo initiator TPO and the obtained PUA, PAHS, BPs in 2-propanol with about 40% total solid content. The UV-curable solution was spin-coated on a plastic substrate, and then followed by UV irradiation (broadband, 400 mj/cm^2^) to obtain a cured PUA/PAHS HN coating.

#### 2.3.2. Physicochemical Properties of Materials

*Characterization:* SEM images were acquired from a Zeiss SUPRATM 55 SAPPHIRE (Oberkochen, Germany) field-emission scanning electron microscope. The SEM images were used to analyze the morphology of materials. The preparation method for samples was to take part of the UV- cured coating and paste it onto the sample table through conductive resin. The cross-section samples were pasted onto the cross-section sample table after freezing extraction. The AFM images were acquired from the Bruker Icon (Karlsruhe, Germany) atomic force microscope and were used to analyze the morphology of materials by detecting the atomic force between the sample and the probe. The samples were dispersed in EtOH and then dropped onto Si substrates for investigation. FTIR spectra were collected in a wavenumber range of 4000–400 cm^−1^ on a Thermo Nicolet IS5 instrument (Waltham, MA, USA). The FTIR spectra were used to analyze the molecular structure of materials through functional group recognition. The preparation method for samples was KBr tableting. Raman scattering was conducted on a Horiba Jobin-Yvon Lab Ram HR VIS high-resolution confocal Raman microscope (Paris, France) equipped with a 633 nm laser. The Raman FTIR spectra were used to analyze the structure of materials by Raman peak recognition. The samples were dispersed in EtOH and then dropped onto Si substrates for investigation. The ^1^H NMR spectroscopy was performed on the Bruker Advance DRX-300 spectrometer (Karlsruhe, Germany) at 25 °C and was used to analyze the structure of materials by NMR peak recognition. The samples were dissolved in deuterium reagent and then put into the nuclear magnetic tube. XPS spectra were obtained from a Thermo Escalab 250Xi spectrometer (Waltham, MA, USA) equipped with an X-ray source producing Al Kα radiation (1486.6 eV). The XPS spectra were used to analyze the surface elements of materials. The samples were dispersed in EtOH and then dropped onto Si substrates for investigation.

#### 2.3.3. Performance Testing of Materials

*Measurement of mechanical properties of coatings:* The adhesion strength of coatings was measured on a BGD500 direct pull tensile force machine by a pull-off test. The pull-off test was classified as a near to surface, partially destructive method that was able to measure the maximum tensile strength of the coatings. The hardness of the UV-cured coatings was measured by industrial pencil hardness tests (JIS K5400) on a QHQ-A pencil hardness tester. The tip of the pencil is placed on the coated substrate and scratched over the film. The hardness designation of the pencil that just fails to cut the film is the pencil hardness of the film. The lubrication performance of coatings was evaluated by coefficient of friction tests. The friction tests were completed using an MS-T3001 ball disc friction tester. A GG15 ball with a diameter of 6 mm was a fixed friction, pair and the coating to be tested is a rotary disc.

The friction coefficient was recorded in real time by the equipment system.

*Measurement of antifogging performance of coatings:* The antifogging performance of coatings was evaluated by three methods. The first method was the hot water vapor test. The samples were held above a water bath containing 60 °C water, and the distance between the samples and water surface was 5 cm. The antifogging performance was measured by observing the fogging of coated substrates. The second method was to measure the hydrophilicity of coatings by testing the WCA. A 6 μL drop of water was placed onto the surface of the coating, and the WCA was analyzed by the image of water spreading. The third method was to investigate the light transmission over the 300–800 nm wavelength range using a UV-Vis spectrophotometer during fogging tests.

*Measurement of antifogging cycle of coatings:* The antifogging cycle tests were carried on a 60 °C water bath. The sample was first exposed to hot water vapor (60 °C) for 15 min (denoted as the wet state). Then, the sample was dried at 40 °C for 4 h (denoted as the dry state). One cycle was completed, and the next cycle was carried out according to this method.

*Measurement of high and low temperature cycling resistance of coating:* Placing the sample in a high and low temperature box, the program was set as follows: cool down to −20 °C, keep the temperature at −20 °C for 12 h, and then raise the temperature to 80 °C, keep the temperature at 80 °C for 12 h, and the rate of temperature rise and fall is 1 °C/min. One cycle was completed, and the next cycle was carried out according to this method. The antifogging performance of the coatings was evaluated after 10 cycles. 

## 3. Results and Discussion

### 3.1. Material Synthesis and Characterization

The polymer HN consisted of polyurethane acrylate (PUA) and acrylic resin (poly (acrylic acid (AA)-2-hydroxyethyl methacrylate (HEMA)-sulfobetaine methacrylate (SBMA), short for PAHS)), prepared by UV-initiated cross-linking. Both polymers have outstanding hydrophilicity and prominent adhesion on plastic substrates [11,30]. The UV-curable six functional PUA was prepared by polymerization of polyethylene glycol (PEG) with diisocyanate and end-capping with pentaerythritol triacrylate (PETA) (Appendix A). PAHS was obtained by radical polymerization of three hydrophilic monomers AA, HEMA, SBMA, which contained generous hydrophilic groups (Appendix A). The ^1^H nuclear magnetic resonance (^1^H NMR) spectra and the Fourier transform infrared (FTIR) spectra of PUA and PAHS illustrated the successful synthesis of the two polymers (Appendix A).

The BP nanosheets (BPs) were obtained by a liquid exfoliation method reported by our group [31]. The scanning electron microscope (SEM) image (Appendix A) shows BPs about 200–300 nm in size. The UV-curable solution prepared by dissolving the as-obtained PUA, PAHS, BPs and photo initiator in solvent was spin-coated on a plastic substrate and irradiated by UV to synthesize the BPs hybrid polymer HN coating (denoted as PUA/PAHS/BPs HN). As shown in Figure 1a, the hybridization of BPs and the cross-linking of PUA and PAHS occurred simultaneously under UV irradiation. The photo initiator produced free radicals to initiate polymerization of acrylate groups and generation of P-C bonds, thus embedding BPs in the polymer HN structure. In addition to the generation of P-C bonds, the interactions between P atoms and hydrophilic groups also enhances the stability of BPs. PUA/PAHS/BPs HN showed not only long-term stability but also high mechanical strength due to the crosslinking of polymers. When water vapor condensed on the surface of PUA/PAHS/BPs HN, a hemi-wicking phenomenon occurred on the hydrophilic rough surface (Figure 1b) [32]. The hybridization of BPs increased the surface roughness, which made the water spread more rapidly on the surface of the coating, thus realizing the effective antifogging performance of PUA/PAHS/BPs HN.

The SEM images of PUA/PAHS HN and PUA/PAHS/BPs HN are shown in Figure 2a,b, respectively. The network structure formed by the cross-linking of PUA and PAHS had a diameter of about 300–500 nm, with micron-scale hole diameter and about 70% porosity. The morphology of the network structure remained basically unchanged with the addition of BPs. The cross-section SEM image of PUA/PAHS/BPs HN in Appendix A shows its good homogeneity and tight combination with the substrates. The AFM diagram of PUA/PAHS/BPs HN showed that BPs were relatively evenly distributed in the coating (Appendix A). The FTIR peaks of PUA at 3000–3100 cm^−1^ and 1650 cm^−1^ were attributed to C–H and C=C, respectively (Figure 2c and Appendix A). Compared with PUA, the C–H and C=C peaks of PUA/PAHS/BPs HN disappeared, and some new peaks such as those at 1039 cm^−1^ and 1240 cm^−1^ associated with the characteristic peak of PAHS and BPs, respectively, appeared. This result shows that the free radical polymerization of C=C and the hybridization of BPs were realized synchronously. High-resolution XPS (HR-XPS) spectra of PUA/PAHS/BPs HN were acquired and analyzed (Figure 2d,e and Appendix A). As shown by the C 1s XPS spectrum, P–C, C–O, and C=O peaks at 284.1, 286.1, and 288.4 eV, respectively, confirmed the existence of P-C bonding and carbon oxygen covalent bonding of the polymers. The P 2p spectrum showed the P 2p_3/2_ and P 2p_1/2_ doublets at 129.6 and 130.5 eV, respectively, characteristic of crystalline BP. In addition, the broad peak at 133.3 eV corresponded to P-C covalent bonds, corroborating chemical binding between BP and PUA by free radical reaction. Compared with PUA/PAHS HN, the Raman spectrum of PUA/PAHS/BPs HN shows three prominent peaks of BPs related to A^1^_g_ at 361 cm^−1^, B_2g_ at 438 cm^−1^ and A^2^_g_ at 466 cm^−1^, respectively [33], indicating preservation of BPs structure during hybridization (Figure 2f). These results imply successful preparation of the HN structure and hybridization of BPs.

### 3.2. Mechanical Properties of Coatings

The mechanical properties of PUA/PAHS HN and PUA/PAHS/BPs HN were further evaluated. Compared with PUA/PAHS HN, the pencil hardness of PUA/PAHS/BPs HN increased from HB to 3H, which illustrates that BPs improved the hardness of the coating (Figure 3a). Meanwhile, the WCA of PUA/PAHS/BPs HN decreased with the increase in BPs content, indicating that the introduction of BPs improved the roughness of the coating. The simultaneous improvement of roughness and hardness through BPs hybridization was beneficial to the scratch resistance of the coating. Since adhesion strength of the coating determines its peeling resistance, pull-off adhesion strength tests of PUA/PAHS/BPs HN and PUA/PAHS HN on polyethylene terephthalate (PET), polycarbonate (PC), polymethyl methacrylate (PMMA) and acrylonitrile-butadiene-styrene terpolymer (ABS) were performed. All substrates were transparent and smooth plates (size 50 mm × 50 mm, thickness 3 mm), and the thickness of the coatings obtained on the different substrates was 50 μm by default. As shown in Figure 3b, the adhesion strength of PUA/PAHS HN was between 1.7–3.5 MPa, illustrating that topological entanglement and covalent bonding with substrates of polymer HN made the coating firmly adhere on different plastic substrates [28]. The introduction of BPs further enhanced the adhesion of the coating, indicating that hybridization of BPs improves the topological entanglement between polymers. The coefficient of friction (COF) continued to decline with the increase in BPs content. Compared with the HN coating without BPs, the COF of PUA/PAHS/BPs HN decreased by 68%, indicating that successful hybridization of BPs greatly improved the lubricity of the coating (Figure 3c). Benefiting from the lubrication of BPs, the WCA of PUA/PAHS/BPs HN did not rise significantly after 1000 friction test cycles, showing its excellent friction resistance property (Figure 3d). These results indicate that the hybridization of BPs can effectively improve the hardness, roughness, adhesion strength and friction resistance of polymer HN.

### 3.3. Antifogging Performances of Coatings

When BPs content of the HN coating was less than 6 wt%, PUA/PAHS/BPs HN had a light transmittance higher than 90% (Appendix A). Considering the transmittance and mechanical properties of the coating, we chose to add 6 wt% BPs into the HN coating thereafter. The sustained antifogging performance of PUA/PAHS/BPs HN was evaluated by the 60 °C hot water vapor test for 60 min. Compared with bare PMMA slide, the PMMA slide coated with PUA/PAHS/BPs HN presented a super-hydrophilic state (WCA = 8°) and had a light transmittance higher than 90% over the 300–800 nm wavelength range under 60 min continuous antifogging test (Figure 4a). The optical photographs in Figure 4b show that the coated PMMA slide did not fog when exposed to hot water vapor (60 °C) for 60 min, while the bare PMMA slide fogged even in the first minute. These results indicate that the BPs hybrid polymer HN coating possesses sustained antifogging ability. In addition, PUA/PAHS/BPs HN on PMMA slides remained highly transparent (optical transmittance over 90%) during seven wet–dry cycles of antifogging tests, illustrating its long-term and stable antifogging performance (Figure 4c). To further analyze the influence of BPs on the antifogging performance of the coating, sustained antifogging tests of PUA/PAHS HN and PUA/PAHS/BPs HN with different thicknesses were conducted. As the increase in coating thickness could delay water invasion, thicker PUA/PAHS HN yielded antifogging performance for a longer duration (Figure 4d). Unfortunately, it was still difficult to maintain 60 min antifogging duration with the PUA/PAHS HN even with increased thickness. In contrast to the PUA/PAHS HN, the PUA/PAHS/BPs HNs with 5 μm, 20 μm and 50 μm thicknesses were able to maintain over 90% light transmittance when exposed to hot water vapor (60 °C) for 60 min. The optical photographs in Figure 4e show that antifogging performance of PUA/PAHS HN declined after being immersed in water for 1 day, and completely failed after 5 days. However, the PUA/PAHS/BPs HN still maintained long-term antifogging ability even after being immersed in water for 5 days. These results demonstrate the outstanding water resistance and long-term antifogging performance of PUA/PAHS/BPs HN. To further study the stability of PUA/PAHS/BPs HN, its hydrophilicity was tested after long-term exposure to humid air. The WCA of PUA/PAHS/BPs HN on different substrates did not rise significantly after exposure to air for 6 weeks (Figure 4f), while the WCA of PUA/PAHS HN rose significantly after exposure to air for 1 week. In addition, PUA/PAHS/BPs HN had better high and low temperature cycling resistance than PUA/PAHS HN (Appendix A), suggesting that the hybridization of BPs enhanced the thermal stability of HN. These results illustrate that the BPs hybrid polymer HN coating has excellent water-resistant and antifogging performance and outstanding stability, thus achieving its long-term antifogging capability.

### 3.4. Long-Term Antifogging Mechanism of Coatings

Based on these results, the long-term antifogging mechanism of PUA/PAHS/BPs HN is proposed in Figure 5a. Due to the different surface tensions of BPs and organic polymers, solvent evaporation drives Bénard Marangoni convection during the process of UV curing [34]. BPs tend to migrate towards the coating surface due to Bénard Marangoni convection, resulting in the generation of microstructure and increased roughness on the surface. Due to excellent water-based lubricity and water absorption capacity, BPs on the surface make it difficult for water to penetrate towards the inside of the coating, preventing antifogging failure induced by water invasion. To verify this assumption, WCA tests, microscopic observations and water penetration tests of PUA/PAHS/BPs HN were conducted. With the increase in UV curing time, the WCA of the HN coating decreased from 23° to 8°, indicating increased roughness of the coatings during UV curing (Figure 5b). Optical microscopic observation of the coating surface (about 1 μm thick) before and after UV curing was further conducted. The microscopic photographs of PUA/PAHS/BPs HN in Figure 5c show that BPs migrated towards the coating surface during UV curing, resulting in enrichment of BPs on the surface. These results prove that Bénard Marangoni convection of BPs and polymers occurred in this study. The improvement of hydrophilicity is attributed to the increase in coating roughness after enrichment of BPs on the surface. To further analyze the water resistance of PUA/PAHS/BPs HN, its water permeability was tested after being immersed in water for 5 days. The water penetration rate of PUA/PAHS/BPs HN decreased with the addition of BPs content and promotion of the mass ratio of PUA to PAHS (Figure 5d). The crosslinking density of polymers increased with the addition of PUA content, resulting in a denser HN structure and lower water penetration rate. In addition, the introduction of BPs contributed to blocking the permeation of water, thus enhancing the water resistance of PUA/PAHS/BPs HN in a humid environment.

The above results demonstrate the long-term antifogging performance mechanism of the proposed coating.

## 4. Conclusions

In summary, for the first time, a BPs hybrid super-hydrophilic polymer HN coating (PUA/PAHS/BPs HN) was prepared by UV curing for long-term antifogging performance. Different from physical blending, covalent P-C bonds between BPs and polymer were generated by UV-initiated free radical reaction, resulting in BPs firmly embedded in the polymer HN structure. The BPs enriched on the coating surface by UV regulating migration can effectively prevent the permeation of water towards the inside of the coating through its own good water-based lubricity and water absorption capacity. Compared with nonhybrid polymer HN, PUA/PAHS/BPs HN not only has higher hardness and better friction resistance properties, but also exhibits superior water resistance and longer antifogging duration. Since water invasion is greatly reduced by BPs, PUA/PAHS/BPs HN maintained 60 min antifogging duration under the 60 °C water vapor test and still maintained long-term antifogging performance after being immersed in water for 5 days. After exposure to air for 6 weeks, the antifogging performance of PUA/PAHS/BPs HN did not decline, showing its outstanding stability. This study provides not only a method for fabricating BP hybrid polymer materials through the generation of P–C bonds induced by free radical reaction, but also a new way for regulating the directional enrichment of BP towards the surface of a composite structure, which presents new approaches for long-term antifogging in humid environments.

## Figures and Tables

**Figure 1 nanomaterials-13-00086-f001:**
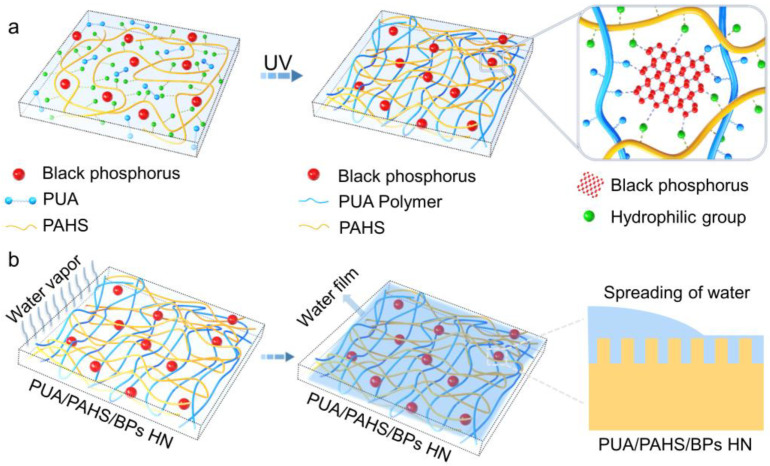
Schematic diagram of design of BP hybrid polymer HN coating for effective antifogging performance. (**a**) Synthesis of PUA/PAHS/BPs HN. (**b**) Antifogging mechanism of PUA/PAHS/BPs HN.

**Figure 2 nanomaterials-13-00086-f002:**
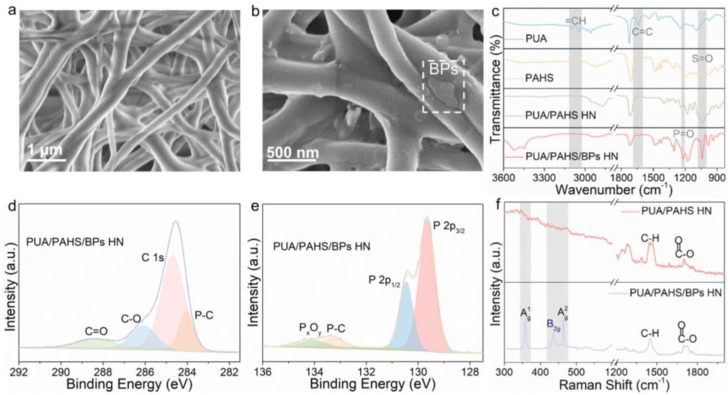
Characterization of BPs hybrid polymer HN coating and nonhybrid polymer HN coating. (**a**) SEM image of PUA/PAHS HN. (**b**) SEM image of PUA/PAHS/BPs HN. (**c**) FTIR spectra of PAHS, PUA, PUA/PAHS HN and PUA/PAHS/BPs HN. (**d**) HR-XPS C 1s spectrum of PUA/PAHS/BPs HN. (**e**) HR-XPS P 2p spectrum of PUA/PAHS/BPs HN. (**f**) Raman spectra of PUA/PAHS HN and PUA/PAHS/BPs HN.

**Figure 3 nanomaterials-13-00086-f003:**
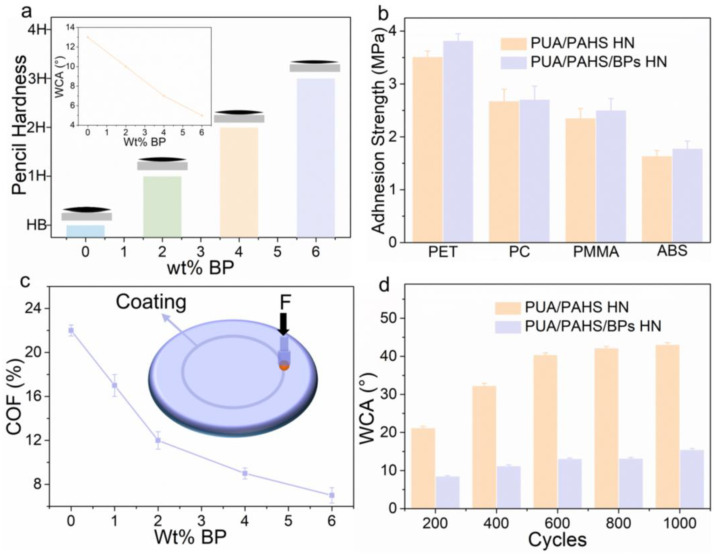
Mechanical properties of BPs hybrid polymer HN coating and nonhybrid polymer HN coating. (**a**) Pencil hardness of PUA/PAHS/BPs HN with different BPs content. Inset: WCA of PUA/PAHS/BPs HN with different BPs content. (**b**) Adhesion strength of PUA/PAHS/BPs HN and PUA/PAHS on PET, PC, PMMA and PBS substrates. (**c**) Coefficient of friction (COF) curve of PUA/PAHS/BPs HN with different BPs content. Inset: schematic diagram of ball disc friction tests. (**d**) WCA of the PUA/PAHS/BPs HN and PUA/PAHS HN under different friction cycle tests.

**Figure 4 nanomaterials-13-00086-f004:**
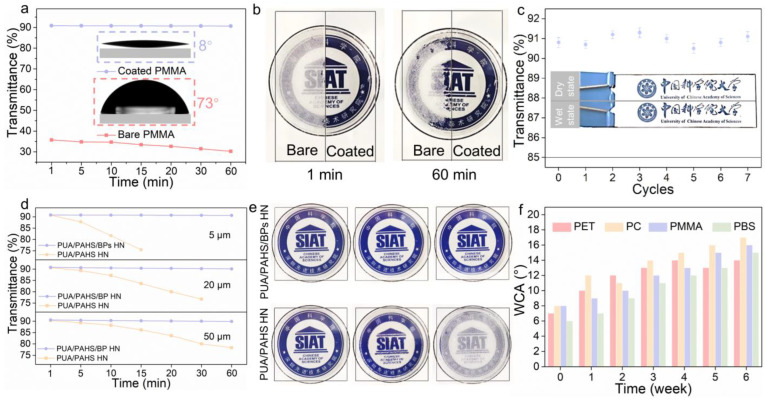
Antifogging performances of BPs hybrid polymer HN coating and nonhybrid polymer HN coating. (**a**) The average transmittance of PUA/PAHS/BPs HN-coated PMMA and bare PMMA (transparent and smooth plates, size 75 × 25 mm, thickness 3 mm) when exposed to hot water vapor (60 °C) for 60 min. Inset: WCA of PUA/PAHS/BPs HN-coated PMMA (8°) and bare PMMA (73°). (**b**) Optical photographs of a bare PMMA slide and a PMMA slide coated with PUA/PAHS/BPs HN when exposed to hot water vapor (60 °C) for 1 min and 60 min. (**c**) The average transmittance of a PMMA slide coated with PUA/PAHS/ BPs HN during cyclic antifogging tests. The optical photographs of the PMMA slide show a good field of vision during antifogging test after seven wet–dry cycles. (**d**) The average transmittance of PMMA slides coated with PUA/PAHS HN and PUA/PAHS/BPs HN, respectively, with varying thickness when exposed to hot water vapor (60 °C) over time. (**e**) Antifogging behavior of PMMA slides (transparent and smooth plates, size 50 mm× 50 mm, thickness 3 mm) coated with PUA/PAHS HN and PUA/PAHS/6 wt% BPs HN, respectively, after being immersed in water for different periods of time. (**f**) WCA of PUA/PAHS/BPs HN on PET, PC, PMMA and PBS substrates after exposure to air for 1, 2, 3, 4, 5 and 6 weeks, respectively.

**Figure 5 nanomaterials-13-00086-f005:**
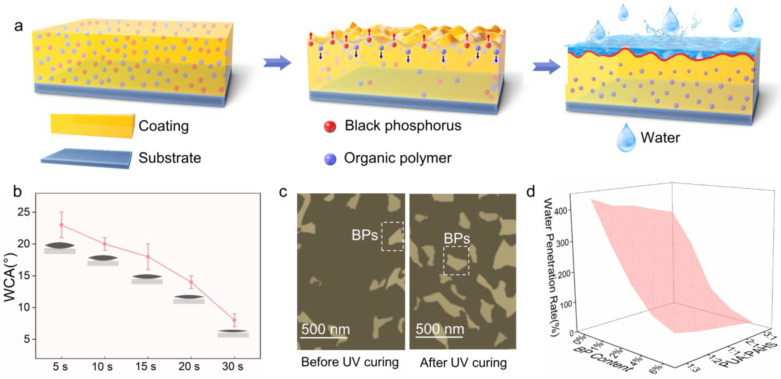
(**a**) Schematic diagram of long-term antifogging mechanism of PUA/PAHS/BPs HN. (**b**) WCA of PUA/PAHS/BPs HN with different UV curing times. (**c**) Microscopic photographs of PUA/PAHS/BPs HN before and after UV curing. (**d**) Water penetration rate of PUA/PAHS/BPs HN with different BPs content and mass ratio of PUA to PAHS.

## Data Availability

The data presented in this study are available on request from the corresponding author.

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
