# Peer review of "Long-Term Antifogging Coating Based on Black Phosphorus Hybrid Super-Hydrophilic Polymer Hetero-Network"

_nanomaterials, 2022, doi:10.3390/nano13010086_

Round 1
Reviewer 1 Report
The authors address the issue of short-term effectiveness of antifogging coating and propose a polymer hetero-network coating for this purpose. The proposed method based on UV-curing allows to achieve long-term water resistance which has been demonstrated in experiments. The proposed technique is of particular interest and can be used for example in manufacturing of certain optical instruments where high light transmittance is an important criteria. The proposed method is based on the already existed strategy of super-hydrophilic polymer usage and improve this method by enhancing its long-term water-resistance. The authors comprehensively describe material preparation and testing and demonstrate good performance of the proposed technique. I am sure that the manuscript can be of great interest for a many researchers involved in both fundamental and applied studies. I believe that the manuscript should be published in “nanomaterials” after addressing of a few comments listed below.
- As the authors have mentioned in introduction, hydrophilicity is one way to achieve antifogging effect although it is (almost) inevitably leads to low water resistance. I believe it would be nice if the authors have mentioned the strategies other than high hydrophilicity which can be used for antifogging purposes.
- As far as I know in some cases stability of certain coating types can be strongly reduced by multi-cyclic decrease and increase of the material temperature. In practical use of many devices such a dynamical temperature changes is not unusual thing, which sometimes lead to coating failure. The authors have performed several measurements of antifogging performance in relation with its mechanical stability, but could you comment on the coating stability in relation with multi-cycle temperature changes.
- According to the figure 4(d) in the case of BP-coating it seems that thickness of the coating layer doesn’t affect durability of the coating at all. Quite dramatic increase of the coating thickness from 5 to 50 um results in no difference in performance under 60 minutes of hot water vapor impact. Do you think that PUA/PAHS/BPs HN coating can be used with the thickness as low as 5 um, or increase of its thickness actually leads to some improvements?
Best regards,
Reviewer
Author Response
Response to Referees for manuscript nanomaterials-2090463
We thank the reviewers for their helpful suggestions and constructive comments. We have addressed all the concerns raised by the reviewers. As a result, the paper has been substantially revised.
Changes to the text are highlighted in the revised manuscript file in blue. In the following, we address the reviewers’ comments one by one.
Reviewer #1:
General comment
The authors address the issue of short-term effectiveness of antifogging coating and propose a polymer hetero-network coating for this purpose. The proposed method based on UV-curing allows to achieve long-term water resistance which has been demonstrated in experiments. The proposed technique is of particular interest and can be used for example in manufacturing of certain optical instruments where high light transmittance is an important criteria. The proposed method is based on the already existed strategy of super-hydrophilic polymer usage and improve this method by enhancing its long-term water-resistance. The authors comprehensively describe material preparation and testing and demonstrate good performance of the proposed technique. I am sure that the manuscript can be of great interest for a many researchers involved in both fundamental and applied studies. I believe that the manuscript should be published in “nanomaterials” after addressing of a few comments listed below.
Our response
We thank the reviewer for appreciating our approach to synthesize a black phosphorus hybrid super-hydrophilic polymer hetero-network coating for long-term antifogging, and for acknowledging the potential of our work.
Comment 1
As the authors have mentioned in introduction, hydrophilicity is one way to achieve antifogging effect although it is (almost) inevitably leads to low water resistance. I believe it would be nice if the authors have mentioned the strategies other than high hydrophilicity which can be used for antifogging purposes.
Our response
We thank the reviewer for the positive comments. Actually, many strategies, referring to super-hydrophobic and super-hydrophilic surfaces, have been developed for effective antifogging. Super-hydrophobic surfaces eliminate the effect of fog droplets by rolling down from the surface, but the light transmission tends to be compromised by its own micro/nano-structured roughness. Different from super-hydrophobic surfaces, the antifogging coating based on super-hydrophilic polymer avoids fogging by quickly spreading fog droplets into a continuous water film, so as to effectively prevent the light scattering. Therefore, in the field of antifogging, super-hydrophilic surfaces are easier, more reliable and more promising than super-hydrophobic surfaces. According to the suggestion, the corresponding introdution and references are added in the manuscript.
Our modification to the manuscript:
(Line 38, page 1: in introduction section of revised main text)
“Many strategies, referring to super-hydrophobic and super-hydrophilic surfaces, have been developed for effective antifogging. Super-hydrophobic surfaces eliminate the effect of fog droplets by rolling down from the surface, but the light transmission tends to be compromised by its own micro/nano-structured roughness [5-7]. Different from super-hydrophobic surfaces, the antifogging coating based on super-hydrophilic polymer avoids fogging by quickly spreading fog droplets into a continuous water film, so as to effectively prevent the light scattering. Therefore, in the field of antifogging, super-hydrophilic surfaces are easier, more reliable and more promising than super-hydrophobic surfaces [8-12].”
(Line 418, page 10: in reference section of revised main text)
- Sun, Z.; Liao, T.; Liu, K.; Jiang, L.; Kim, J. H.; Dou, S. X. Fly-eye inspired superhydrophobic anti-fogging inorganic nanostructures. Small 2014, 10, 3001-3006. https://doi.org/10.1002/smll.201400516.
- Darband, G. B.; Aliofkhazraei, M.; Khorsand, S.; Sokhanvar, S.; Kaboli, A. Science and engineering of superhydrophobic surfaces: review of corrosion resistance, chemical and mechanical stability. Arabian J. Chem. 2020, 13, 1763-1802. https://doi.org/10.1016/j.arabjc.2018.01.013.
- Mouterde, T.; Lehoucq, G.; Xavier, S.; Checco, A.; Black, C. T.; Rahman, A.; Midavaine, T.; Clanet, C.; Quere, D. Antifogging abilities of model nanotextures. Nat. Mater. 2017, 16, 658-663. https://doi.org/10.1038/NMAT4868.
Comment 2
As far as I know in some cases stability of certain coating types can be strongly reduced by multi-cyclic decrease and increase of the material temperature. In practical use of many devices such a dynamical temperature changes is not unusual thing, which sometimes lead to coating failure. The authors have performed several measurements of antifogging performance in relation with its mechanical stability, but could you comment on the coating stability in relation with multi-cycle temperature changes.
Our response
Thank you for the comment. This is a very valid point. It is true that the stability of certain coating types can be strongly reduced by multi-cyclic decrease and increase of the material temperature. We have added the high and low temperature (-20-80 oC) cycle tests of PUA/PAHS HN and PUA/PAHS/BPs HN. The antifogging performance of PUA/PAHS HN declines after multiple cycles while PUA/PAHS/BPs HN keeps excellent antifogging performance after 10 cycles. We believe that the hybridization of black phosphorus improves the thermal stability of the coating. The corresponding results and discussions are added in the manuscript.
Our modification to the manuscript:
(Line 179, page 4: in materials and methods section of revised main text)
“Measurement of high and low temperature cycling resistance of coating: place the sample in high and low temperature box, the program is set as follows: cool down to -20 oC, keep the temperature at -20 oC for 12 h, and then raise the temperature to 80 oC, keep the temperature at 80 oC for 12 h, and the rate of temperature rise and fall is 1 oC/min. One cycle was completed, and the next cycle was carried out according to this method. The antifogging performance of coatings was evaluated after 10 cycles.”
(Line 315, page 7: in results and discussion section of revised main text)
“In addition, PUA/PAHS/BPs HN has better high and low temperature cycling resistance than PUA/PAHS HN (Figure S12), suggesting that the hybridization of BPs enhances the thermal stability of HN.”
(Figure S12: in revised Supporting information)
Figure S12. The average transmittance of a PMMA slide coated with PUA/PAHS/ BPs HN during antifogging tests after high and low temperature cycles.
Comment 3
According to the figure 4(d) in the case of BP-coating it seems that thickness of the coating layer doesn’t affect durability of the coating at all. Quite dramatic increase of the coating thickness from 5 to 50 um results in no difference in performance under 60 minutes of hot water vapor impact. Do you think that PUA/PAHS/BPs HN coating can be used with the thickness as low as 5 um, or increase of its thickness actually leads to some improvements?
Our response
Thank you for the comment. We think the increase thickness of PUA/PAHS/BPs HN may leads to some improvements. Generally, the hydrophilic components of the coating increase with the increase of the coating thickness, which is beneficial to antifogging. On the other hand, when the coating increases to a certain thickness, the effect of water invasion is relatively small. Therefore, increase thickness of PUA/PAHS/BPs HN may leads to improvement of antifogging performance. Actually, the coating with 50 um thickness shows better performance than the coating with 5 um thickness during more than 60 minutes of hot water vapor impact.

Reviewer 2 Report
The manuscript by Lie Wu et al. entitled “Long-term antifogging coating based on black phosphorous hybrid super-hydrophilic polymer hetero-network”, presents an experimental work on the production of antifogging coatings for polymers. This is an interesting work that may lead to applications in real life situations. Here are the main aspects that need improvement prior to acceptance:
1. Please give full details about the substrates used to obtain the coatings (size of samples, thickness, roughness, …).
2. Please give full details about the different analytical techniques used to evaluate the physicochemical properties of materials. Just a list of the techniques employed is not enough information. Same happens with the measurement of the mechanical properties of the coatings: please give full information about the methods and equipment used.
3. Please give information about the thickness of the coatings obtained on the different substrates.
4. Please give cross-section images of coatings in order to show the homogeneity of these coatings and also the interface with the substrates.
Author Response
Response to Referees for manuscript nanomaterials-2090463
We thank the reviewers for their helpful suggestions and constructive comments. We have addressed all the concerns raised by the reviewers. As a result, the paper has been substantially revised.
Changes to the text are highlighted in the revised manuscript file in blue. In the following, we address the reviewers’ comments one by one.
Reviewer #2:
General comment
The manuscript by Lie Wu et al. entitled “Long-term antifogging coating based on black phosphorous hybrid super-hydrophilic polymer hetero-network”, presents an experimental work on the production of antifogging coatings for polymers. This is an interesting work that may lead to applications in real life situations. Here are the main aspects that need improvement prior to acceptance:
Our response
We thank the reviewer for his positive evaluation of our work and for his comments the addressing of which contributed to the improvement of our manuscript.
Comment 1
Please give full details about the substrates used to obtain the coatings (size of samples, thickness, roughness, …).
Our response
We thank the reviewer for the suggestion. The full details about the substrates are given in the revised main text.
Our modification to the manuscript:
(Line 262, page 6: in results and discussion section of revised main text)
“All substrates are transparent and smooth plates (size 50x50 mm, thickness 3 mm).”
(Line 324, page 8: in results and discussion section of revised main text)
“(transparent and smooth plates, size 75x25 mm, thickness 3 mm)”
(Line 332, page 8: in results and discussion section of revised main text)
“(transparent and smooth plates, size 50x50 mm, thickness 3 mm)”
Comment 2
Please give full details about the different analytical techniques used to evaluate the physicochemical properties of materials. Just a list of the techniques employed is not enough information. Same happens with the measurement of the mechanical properties of the coatings: please give full information about the methods and equipment used.
Our response
Thank you for the suggestion. The full details about the different analytical techniques and full information about the measurement of the mechanical properties of the coatings are given in the revised main text.
Our modification to the manuscript:
(Line 129, page 3: in materials and methods section of revised main text)
“2.3.2 Physicochemical properties of materials
Characterization: The SEM images were acquired from the Zeiss SUPRATM 55 SAPPHIRE field-emission scanning electron microscope. The SEM images were used to analyze the morphology of materials. The preparation method of samples was to take part of the UV cured coating and paste it onto the sample table through conductive resin. The cross-section samples were pasted onto the cross-section sample table after freezing extraction. The AFM images were acquired from the Bruker Icon atomic force microscope and were used to analyze the morphology of materials by detecting the atomic force between the sample and the probe. The samples were dispersed in EtOH and then dropped onto Si substrates for investigation. The FTIR spectra were collected in wavenumber range of 4000-400 cm-1 on a Thermo Nicolet IS5 instrument. The FTIR spectra were used to analyze the molecular structure of materials through functional group recognition. The preparation method of samples was KBr tableting. The Raman scattering was conducted on a Horiba Jobin-Yvon Lab Ram HR VIS high-resolution confocal Raman microscope equipped with a 633 nm laser. The Raman FTIR spectra were used to analyze the structure of materials by Raman peaks recognition. The samples were dispersed in EtOH and then dropped onto Si substrates for investigation. The 1H NMR spectroscopy was performed on the Bruker Advance DRX-300 spectrometer (Bruker, Germany) at 25 °C and was used to analyze the structure of materials by NMR peaks recognition. The samples were dissolved in the deuterium reagent and then put into the nuclear magnetic tube. The XPS spectra were obtained from a Thermo Escalab 250Xi spectrometer equipped with an X-ray source producing Al Kα radiation (1486.6 eV). The XPS spectra were used to analyze the surface element of materials. The samples were dispersed in EtOH and then dropped onto Si substrates for investigation.”
(Line 154, page 4: in materials and methods section of revised main text)
“Measurement of mechanical properties of coatings: The adhesion strength of coatings was measured on a BGD500 direct pull tensile force machine by a pull-off test. The pull-off test was classified as a near to surface, partially destructive method which is able to measure maximum tensile strength of coatings. The hardness of the UV cured coatings was measured by the industrial pencil hardness tests (JIS K5400) on a QHQ-A pencil hardness tester. Place the tip of the pencil on the coated substrate and scratch the pencil. The hardness designation of the pencil that just fails to cut the film is the pencil hardness of the film. The lubrication performance of coatings was evaluated by the coefficient of friction tests. The friction tests were completed by a MS-T3001 ball disc friction tester. The GG15 ball with a diameter of 6 mm is a fixed friction pair and the coating to be tested is a rotary disc. The friction coefficient is recorded in real time by the equipment system.”
Comment 3
Please give information about the thickness of the coatings obtained on the different substrates.
Our response
Thank you for the comment. The information about the thickness of the coatings obtained on the different substrates are given in the revised main text.
Our modification to the manuscript:
(Line 263, page 6: in results and discussion section of revised main text)
“and the thickness of the coatings obtained on the different substrates is 50 μm by default.”
Comment 4
Please give cross-section images of coatings in order to show the homogeneity of these coatings and also the interface with the substrates.
Our response
Thank you for the comment. We have added the cross-section image of PUA/PAHS/BPs HN in the Supporting information and related discussion in revised main text.
Our modification to the manuscript:
(Line 224, page 5: in results and discussion section of revised main text)
The cross-section SEM image of PUA/PAHS/BPs HN in Figure S7 shows its good homogeneity and tight combination with the substrates.
(Figure S7: in revised Supporting information)
Figure S7. The cross-section SEM image of PUA/PAHS/BPs HN.

Round 2
Reviewer 2 Report
Authors answered to all queries posed by this reviewer. The manuscript can be accepted for publication.